# Mechanism of anion exchange and small-molecule inhibition of pendrin

Lie Wang [1,6], Anthony Hoang [1,6], Eva Gil-Iturbe [2,3], Arthur Laganowsky [4], Matthias Quick [2,3,5] ✉ & Ming Zhou [1] ✉

Pendrin (SLC26A4) is an anion exchanger that mediates bicarbonate ($HCO_3^-$) exchange for chloride ($Cl^-$) and is crucial for maintaining pH and salt homeostasis in the kidney, lung, and cochlea. Pendrin also exports iodide ($I^-$) in the thyroid gland. Pendrin mutations in humans lead to Pendred syndrome, causing hearing loss and goiter. Inhibition of pendrin is a validated approach for attenuating airway hyperresponsiveness in asthma and for treating hypertension. However, the mechanism of anion exchange and its inhibition by drugs remains poorly understood. We applied cryo-electron microscopy to determine structures of pendrin from *Sus scrofa* in the presence of either $Cl^-$, $I^-$, $HCO_3^-$ or in the apo-state. The structures reveal two anion-binding sites in each protomer, and functional analyses show both sites are involved in anion exchange. The structures also show interactions between the Sulfate Transporter and Anti-Sigma factor antagonist (STAS) and transmembrane domains, and mutational studies suggest a regulatory role. We also determine the structure of pendrin in a complex with niflumic acid (NFA), which uncovers a mechanism of inhibition by competing with anion binding and impeding the structural changes necessary for anion exchange. These results reveal directions for understanding the mechanisms of anion selectivity and exchange and their regulations by the STAS domain. This work also establishes a foundation for analyzing the pathophysiology of mutations associated with Pendred syndrome.

Pendrin is an anion transporter that exports $I^-$ or $HCO_3^-$ in exchange for $Cl^-$, and is important for maintaining the function of epithelial cells in the thyroid gland, inner ear, kidney and lung[1,2]. In humans, mutations in pendrin are known to cause two major hearing disorders: enlarged vestibular aqueduct syndrome (EVAS) and Pendred syndrome, both of which lead to bilateral congenital deafness, with the latter also accompanied by goiter[3,4]. Hearing loss is often progressive during childhood and is accompanied by structural changes to the inner ear including widened vestibular aqueducts, enlarged endolymphatic sacs, and abnormalities in the cochlea[5]. However, the precise mechanism of hearing loss caused by pendrin mutations remains unclear, and elucidating pendrin's structure and function will lead to a better understanding of the disease mechanisms.

Pendrin is also a potential drug target for diuretics and for reduction of lung inflammation[6,7]. Pendrin is overexpressed in lungs affected by chronic obstructive pulmonary disease (COPD), *Bordetella pertussis* infection, cystic fibrosis, and rhinovirus infection[8–10], and linked to lipopolysaccharide-induced acute lung injury. Inhibition of

[1]Verna and Marrs McLean Department of Biochemistry and Molecular Pharmacology, Baylor College of Medicine, Houston, TX, USA. [2]Department of Psychiatry, Columbia University Irving Medical Center, New York, NY, USA. [3]Department of Physiology and Cellular Biophysics, Columbia University Irving Medical Center, New York, NY, USA. [4]Department of Chemistry, Texas A&M University, College Station, TX, USA. [5]Area Neuroscience - Molecular Therapeutics, New York State Psychiatric Institute, New York, NY, USA. [6]These authors contributed equally: Lie Wang, Anthony Hoang. ✉e-mail: mq2102@cumc.columbia.edu; mzhou@bcm.edu

pendrin by small molecules attenuates lung injury in animal studies[11–13]. Inhibitors of pendrin are also known to potentiate the diuretic effects of furosemide in mouse models[7], suggesting a potential way to address the issue of adaptive responses to current diuretics. However, known drugs that target pendrin, such as niflumic acid (NFA), an FDA-approved nonsteroidal anti-inflammatory cyclooxygenase-2 inhibitor, exhibit half-maximal inhibitory concentrations (IC50) in the μM range[7,13,14]. Progress in developing more efficacious inhibitors for pendrin is hindered by the lack of high-resolution structural information of pendrin in a functional context.

Pendrin (SLC26A4) belongs to the SLC26 family of anion transporters, which has 11 members (SLC26A1-11) that mediate transport of inorganic anions such as $SO_4^{2-}$, $Cl^-$, $HCO_3^-$, or $I^-$ ([15,16]). Structures of SLC26A9[17,18], which is a chloride ion channel, SLC26A6 which acts as an electroneutral $HCO_3^-/Cl^-$ and electrogenic oxalate/$Cl^-$ exchanger[19], prestin (SLC26A5)[20–22] which converts mechanical force or motion of the membrane to electrical currents, and the mouse homolog of pendrin[23] were reported previously. Structures of bacterial and plant homologs of SLC26 were also reported[24–26]. These structures reveal conserved features such as a homodimeric assembly, a single anion binding site in the transmembrane domain composed of 14 transmembrane helices, and a C-terminal cytosolic domain termed the Sulfate Transporter and Anti-Sigma factor antagonist (STAS) domain. Structures of SLC26 proteins show that the STAS domain mediates dimerization, but it is not clear if it has any functional roles pertaining to anion transport.

Here, we present structures of pendrin from *Sus scrofa* (ssPendrin), which is 90% identical to human pendrin, in complex with either $I^-$, $HCO_3^-$, or $Cl^-$; ssPendrin in the apo-state; and ssPendrin in the presence of NFA, which inhibits anion transport. These structures, in

conjunction with functional information, reveal insights into the mechanisms of anion transport and inhibition.

## Results

### Anion transport by pendrin

ssPendrin was expressed, purified, and reconstituted into liposomes for anion uptake assays (Supplementary Fig. 1 and Methods). We first examined the transport of $HCO_3^-$ and $I^-$ using radiolabeled compounds (Fig. 1a). Proteoliposomes containing ssPendrin accumulated $^{14}C$-$HCO_3^-$ in time-dependent fashion, reaching a steady-state level of transport after ~10 minutes, whereas control liposomes without pendrin displayed negligible uptake of $^{14}C$-$HCO_3^-$ (Fig. 1b). The uptake of $^{125}I^-$ displayed faster kinetics, reaching a maximum within ~1 min before leveling off to a steady state at ~15 min (Fig. 1c). In support of pendrin-mediated $HCO_3^-/Cl^-$ exchange, we noticed that including 150 mM $Cl^-$ in the ssPendrin-containing proteoliposomes stimulated $HCO_3^-$ (Fig. 1d) or $I^-$ (Fig. 1e) uptake by more than two orders of magnitude, thus raising the possibility that the exchange of anions in pendrin is thermodynamically coupled. We further measured uptake of $^{14}C$-$HCO_3^-$ into ssPendrin proteoliposomes with 1 mM $Cl^-$ outside and varying concentrations of $Cl^-$ inside (Supplementary Fig. 2a). The steady-state amount of uptake after 10 minutes is progressively higher as the $[Cl^-]$ gradient increases, consistent with coupled exchange of anions. Figure 1f shows the effect of different anions included in ssPendrin-containing proteoliposomes on the initial rates of $^{14}C$-$HCO_3^-$ uptake, revealing that the transport of external $HCO_3^-$ is affected differently by the anion species on the *trans* side of the membrane ($Br^- > Cl^- > F^- > NO_3^- > SCN^- > I^-$) (Fig. 1f, Supplementary Fig. 2b). Uptake of $^{14}C$-$HCO_3^-$ was negligible when the proteoliposomes contained the gluconate anion or $SO_4^{2-}$. Taking advantage of the experimentally more tractable

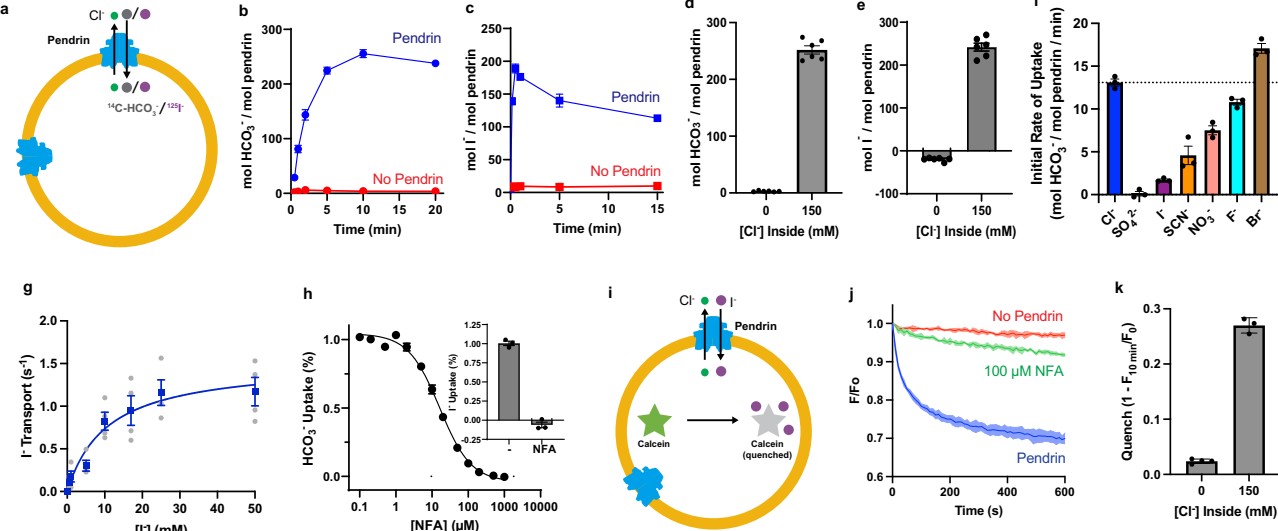

**Fig. 1 | Functional characterization of pendrin. a** Scheme of bicarbonate ($^{14}C$-$HCO_3^-$) or iodide ($^{125}I^-$) uptake into proteoliposomes. **b** Time-dependent accumulation of 100 μM $^{14}C$-$HCO_3^-$ in ssPendrin-containing proteoliposomes (blue) and in liposomes without ssPendrin (red). **c** Time-dependent accumulation of 10 μM $^{125}I^-$ in ssPendrin-containing proteoliposomes (blue) and in liposomes without ssPendrin (red). **d** Accumulation of 100 μM $^{14}C$-$HCO_3^-$ after 10 min with 0 mM or 150 mM $Cl^-$ inside of the proteoliposomes. **e** Accumulation of 10 μM $^{125}I^-$ after 1 min with 0 mM or 150 mM $Cl^-$ inside of the proteoliposomes. **f** Initial rate of accumulation of 100 μM $^{14}C$-$HCO_3^-$ in ssPendrin proteoliposomes with 1 mM anion outside and 100 mM anion inside. The dashed line is the mean of the $Cl^-$ group, y = 13.1. **g** Kinetics of ssPendrin-mediated $I^-$ transport in proteoliposomes. Initial rates of transport were measured with increasing concentrations of $^{125}I^-$. Data (mean ± S.E.M, n = 4) were subject to non-linear regression fitting, yielding a $K_m$ of

10.67 ± 3.46 mM and a $k_{cat}$ of 1.52 ± 0.18 s[-1]. **h** Dose-response curve of NFA inhibition of 100 μM $^{14}C$-$HCO_3^-$ accumulation by ssPendrin after 10 min. Data points were normalized and fit to the equation Y=Bottom + (Top-Bottom)/(1 + (X/IC50)). Insert: Inhibition of 1-min uptake of 10 μM $^{125}I^-$ in ssPendrin-containing proteoliposomes in the presence or absence of 100 μM NFA. **i.** Scheme of $I^-$ uptake measured by collisional quench of calcein fluorescence in proteoliposomes. **j.** Time-dependent transport of 10 mM $I^-$ into proteoliposomes with (green) or without (blue) 100 μM NFA, and into liposomes without ssPendrin (red). **k.** Quench of fluorescence after 10 min with 0 mM or 150 mM $Cl^-$ inside of the proteoliposomes. For all experiments in Fig. 1, each data point is shown as mean ± s.e.m. (n ≥ 3). Lines in Fig. 1b, c simply connect the data points, while the solid line and the shaded areas in Fig. 1j are mean and s.e.m., respectively.

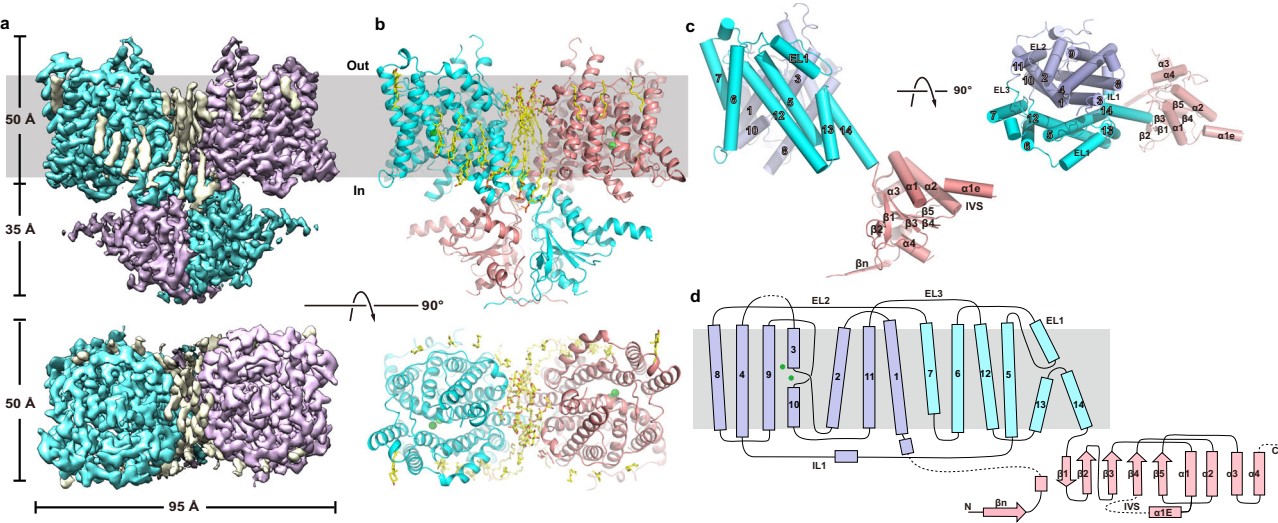

**Fig. 2 | Overall structure of pendrin. a, b** Cryo-EM density map (left) and cartoon representation (right) of the pendrin dimer as viewed from within the plane of the membrane (upper row), or the extracellular side of the membrane (bottom row). The two protomers are colored as cyan and magenta, with lipids in yellow. **c.** Cartoon representation of a pendrin monomer in two orientations. The transport domain, scaffold domain, and STAS domain are colored violet, cyan, and salmon; respectively. **d.** Topology of pendrin. Bound Cl⁻ is shown as green spheres.

[125]I⁻ as ssPendrin substrate, we determined the kinetics of I⁻ import into proteoliposomes that contained 100 mM of internal Cl⁻ (Fig. 1g). ssPendrin exhibited saturable I⁻ transport with a Michalis-Menten constant ($K_m$) of transport of $10.67 \pm 3.46$ mM and a $V_{max}$ (maximum velocity of transport) of $176.82 \pm 20.38$ nmol x µg⁻¹ x min⁻¹, resulting in a catalytic turnover number ($k_{cat}$) of $1.52 \pm 0.18$ s⁻¹ for I⁻ transport, a number comparable to other secondary transporters[27]. NFA, at a concentration of 100 µM abolished the transport of $HCO_3^-$ and I⁻ (Fig. 1h), and we determined the concentration at which NFA inhibited ssPendrin-mediated $HCO_3^-$ transport by 50% (IC50) to be $15.5 \pm 1.5$ µM (Fig. 1h), which is consistent with a previous report[14].

We also examined I⁻ transport by ssPendrin by monitoring collisional quench of a fluorescent dye (calcein) incorporated into the proteoliposomes (Fig. 1i). Significant quench of calcein fluorescence was observed in liposomes reconstituted with pendrin, indicating uptake of iodide, and the uptake is inhibited in the presence of NFA (Fig. 1j). Uptake of I⁻ is also enhanced by the presence of a Cl⁻ concentration gradient (Fig. 1k). Combined, these results demonstrate that ssPendrin is both a $HCO_3^-$/Cl⁻ and I⁻/Cl⁻ exchanger, which is consistent with previous reports describing the functional features of human pendrin[1,2,28].

## Overall structure

The structures of ssPendrin in the presence of I⁻, $HCO_3^-$ or Cl⁻ were determined by cryo-electron microscopy (cryo-EM) to an overall resolution of 2.8 Å, 2.7 Å, and 2.5 Å, respectively (Methods, Supplementary Figs. 3–5, 8). Since the three structures have only minor differences, we describe the Cl⁻-bound ssPendrin here (Fig. 2a, b). The final structural model includes residues 17 to 39, 64 to 585, and 654 to 732, which resolves all 14 transmembrane helices and their connecting loops and most of the STAS domain. Residues 1–16 and 40–63, which are part of the N-terminus preceding the first transmembrane helix (TM); residues 586–653, which are part of the STAS domain and also known as the intervening sequence (IVS)[29]; and residues 733–780, which are the C-terminus after the STAS domain, are unresolved.

ssPendrin forms a homodimer, with each protomer containing 14 transmembrane helices (TM1-14) followed by an intracellular C-terminal STAS domain. The membrane-embedded part of pendrin assumes a structural fold shared by the SLC26 proteins[17–26], SLC4 bicarbonate transporters[30–39] and SLC23 nucleobase transporters[40–43]. TM1-14 fold into two distinct domains: the transport and scaffold domains (Fig. 2c,d). The structural fold has an internal inverted repeat, in which TM1-7 is related to TM8-14 by a pseudo two-fold rotational symmetry (Supplementary Fig. 9). The larger transport domain, which is also called the core domain in previous publications[17–26,30–43], consisting of helices TM1-4 and 8–11, contains the anion binding sites lined by elements from TMs1, 3, 8 and 10. The C-terminal halves of TM3 and TM10 assume well-defined α-helical secondary structures while their N-terminal halves are extended strands. TM3 and TM10 cross roughly in the center of the membrane, and together with residues from TM1 and TM8, form a cradle for substrate anions (Fig. 3d). The two half-helices position the positive ends of their helix dipoles toward the bound substrate, with one anion binding site appearing in all previously reported structures[17–26,30–43].

The smaller scaffold domain, which is also called the gate domain, is composed of TM5-7 and TM12–14 and forms a panel-like structure that contacts the transport domain near the extracellular side. TM13 and 14 of the scaffold domain are significantly shorter, reaching only halfway through the membrane. Compensating for this shortage, EL1, an extracellular loop between TM5 and 6, extends halfway into the membrane (Fig. 2d). The intracellular STAS domain starts after TM14 and extends towards the neighboring protomer without interacting with the TM domain from the same protomer. The dimer interface involves interactions mainly between the two STAS domains, which are swapped between the two subunits.

## Two anion binding sites

Although anions are known to be more susceptible to radiation damage by the electrons and thus contribute less to the density map, we observe two non-protein densities at the crossover region in all three structures, which we assign as S1 and S2 anion binding sites[44] (Fig. 3a–c). The structure of ssPendrin in the presence of gluconate was also determined to an overall resolution of 2.5 Å, in which the S1 and S2 densities are not present, and we call this the apo-state (Fig. 3d, Supplementary Fig. 6). The S1 site has been observed in previous structures of SLC26 proteins[17–26], but the S2 site has not been reported previously. In each structure, densities at the S1 and S2 sites have similar levels. The densities at the S1 and S2 sites are at ~5σ level in both the Cl⁻- and $HCO_3^-$-bound ssPendrin structures, and the densities are at

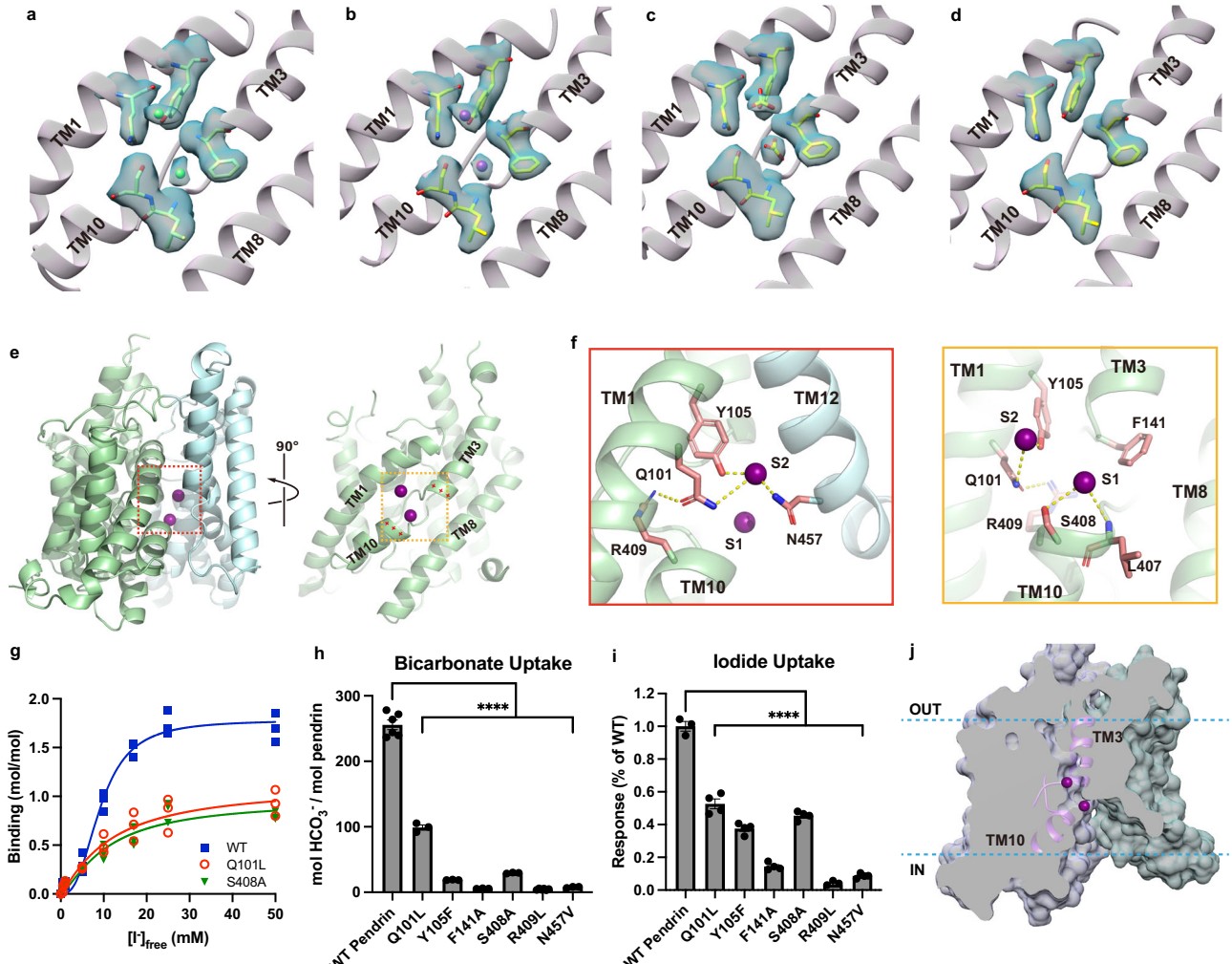

**Fig. 3 | Two anion binding sites. a–d** Anion binding sites of ssPendrin in the presence of Cl⁻ **a**, I⁻ **b**, HCO₃⁻ **c**, and apo **d**, respectively. Residues are shown as sticks, anions as spheres, and densities as cyan surface. **e** Left panel, two anion binding sites (violet spheres) in a pendrin monomer with the transport and scaffold domains colored in green and cyan, respectively. Right panel, anion binding sites and the transport domain. **f** Zoom-in views of the two anion binding sites. Direct interactions are marked with dashed lines. **g** The stoichiometry of ssPendrin anion binding. $^{125}$I⁻ binding by ssPendrin-wild-type (WT), -S408A, and -Q101L was assayed with the SPA. Data ($n = 3$), expressed as mol-to-mol binding ratios, were subjected to the Hill equation in Prism 8, yielding the following $EC_{50}$, Hill coefficient, and molar I⁻-to-protein binding ratios: WT – 9.25 ± 0.49 mM, 2.69 ± 0.34, 1.78 ± 0.06 l; S408A – 10.87 ± 2.66 mM, 1.34 ± 0.31, 0.95 ± 0.12; Q101L – 11.54 ± 3.73, 1.18 ± 0.29, 1.12 ± 0.17. **h, i** 10-min uptake of 100 μM $^{14}$C-HCO₃⁻ and 10 mM I⁻ by ssPendrin-WT or variants with mutations of anion binding site residues. Data points are the mean ± s.e.m. ($n ≥ 3$). Two-tailed Student's $t$ tests were applied for comparison. **** indicates $p < 0.0001$. **j** Cutaway surface representation of pendrin.

~13 σ in the I⁻-bound structure. As a comparison, residue Phe141 in the S1 site and residue Tyr105 in the S2 site have a contour level of ~13 and ~14 σ, respectively, in all three structures.

To functionally corroborate our structural observation of two anion binding sites, we performed $^{125}$I⁻ saturation binding by using the scintillation proximity assay (SPA)[45]. Using known amounts of purified protein allowed us to transform the obtained binding data into molar binding ratios[46]. The I⁻ binding isotherms for ssPendrin-wild-type (WT), the S1 site mutant ssPendrin-S408A, and the S2 site mutant ssPendrin-Q101L plateau at I⁻ concentrations ≥ 25 mM, indicating that saturation equilibrium binding can be achieved at the tested I⁻ concentration range in our experiments (Fig. 3g). Nonlinear regression fitting of the data was performed to obtain the $EC_{50}$, the Hill coefficient, and molar ratios of I⁻-to-ssPendrin binding, revealing that one molecule of ssPendrin-WT can simultaneously bind two I⁻ with an $EC_{50}$ of ~10 mM, and a Hill coefficient of ~2. In contrast, both tested ssPendrin mutants with either impaired S1 or S2 anion binding site exhibited a molar binding stoichiometry of ~1. Whereas the $EC_{50}$ for both mutants were comparable to that of the WT, the Hill coefficients for both mutant

variants were reduced to ~1, indicative of the loss of allosteric binding between the two identified anion sites.

The overall position and coordination of the two anions are almost identical in all three structures, with subtle adjustments of coordinating atoms (Fig. 3a–c). The anion in S1 is coordinated by the backbone hydrogens of Leu407 and Ser408 and the side chain hydroxyl hydrogen of Ser408 (Fig. 3e, f). The side of the aromatic ring of Phe141 from TM3 is within 4.0 Å of S1 and could stabilize the anion in S1 by anion-π interactions[47]. The anion at the S2 position is coordinated by the side-chain hydroxyl hydrogen of Tyr105 and sidechain amide hydrogens of Gln101 and Asn457 (Fig. 3f). While the anion in S1 is coordinated within the transport domain, the anion of S2 involves residue Asn457, which is on TM12 from the scaffold domain. In addition, two elements from the structure enhances the overall positive electrostatic potential at the crossover region. First, the two positive helical dipole moments produced by TM3 and TM10 are focused toward the crossover region, although the S2 site is slightly off the center of the two dipole moments. Second, a highly conserved positively-charged residue, Arg409, is in the crossover region ~7 Å away from either the S1 or S2 anions. The

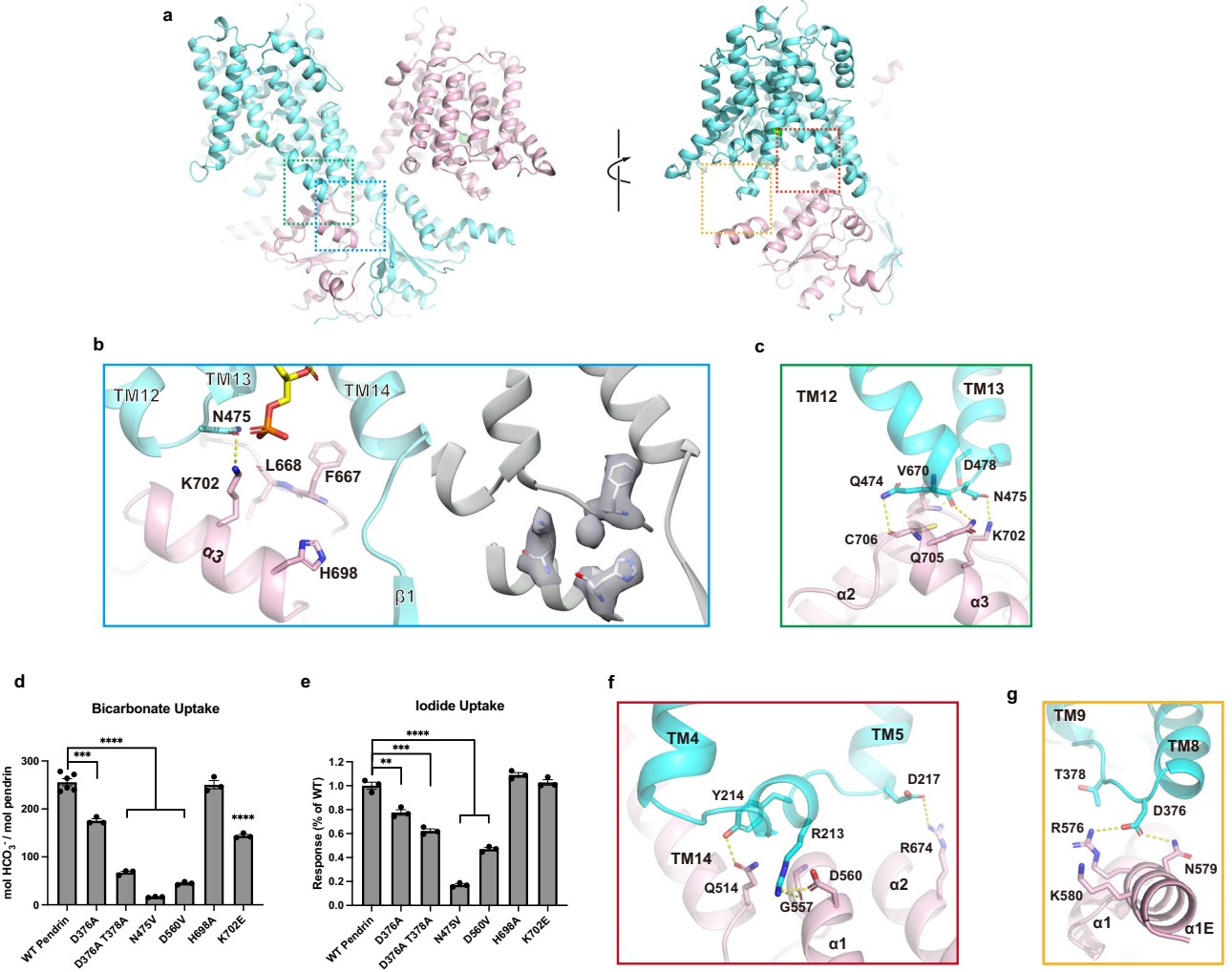

**Fig. 4 | TM-STAS Interactions. a** Overall structure of ssPendrin (PDB ID: 8SGW) in two views. **b** View of interface between the scaffold and STAS domains. **c** View of interface between the transport domain and IVS. **d, e** 10-min uptake of 100 μM $^{14}$C-HCO$_3^-$ or 10 mM I$^-$ by ssPendrin-WT and variants with mutations in interface residues. Each data point is the mean ± s.e.m. (*n* ≥ 3). Two-tailed Student's *t* tests were applied for comparison, **$p < 0.01$; ***$p < 0.001$; and ****$p < 0.0001$. For 4d, D376A, $p = 0.0002$; H698A, $p = 0.6764$. For 4e, D376A, $p = 0.004$; D376A T378A, $p = 0.0004$; H698A, $p = 0.0635$; K702E, $p = 0.5084$. **f** View of interface between the transport and STAS domains. **g** View of interface between scaffold and STAS domains, including density maps for bound anion and surrounding residues.

importance of the positive charge from Arg409 is further highlighted by the naturally occurring R409C/P/H mutations that are known to cause Pendred syndrome[48]. A previous study showed that R409H mutant folds and traffics to the membrane properly[49].

Additional binding site mutations were produced to examine the contribution of these residues to anion transport (Supplementary Fig. 1, Fig. 3h,i). Mutations to residues directly coordinating S1 (F141A and S408A) and S2 (Q101L, Y105F, and N457V) reduce transport significantly, and R409L also significantly reduces transport. As shown in Fig. 3g, mutations S408A and Q101L impair binding to the S1 and S2 binding sites, respectively. However, eliminating one of the two identified anion binding sites does not result in the complete loss of transport, suggesting that a single binding site may be sufficient for ssPendrin-mediated anion translocation. Interestingly, several Pendred syndrome-associated mutations are found in S2, including Y105C[50,51], F141S[52], and N457V[49], and one, S408F[49], found in S1.

In all three ssPendrin structures, the anion binding sites are solvent accessible from the intracellular side through a large cavity formed between the transport and scaffold domains (Fig. 3j). Interactions between the two domains seal off the cavity from the extracellular side. Thus, the structures of ssPendrin are captured in the inward-facing conformation.

## The STAS domain

The cytosolic STAS domain is comprised of five β-strands (β1-5) sandwiched between four α-helices (α1-4). The two STAS domains form a domain-swapped dimer, with a buried surface area of 70 Å$^2$ (Supplementary Fig. 10a, b). Also contributing to the dimer interface is part of the N-terminus, residues 19 to 25 (βn). The two βns from the neighboring protomer form an antiparallel beta sheet that interacts with the dimeric STAS domain (Supplementary Fig. 10c). Structures of different members of SLC26 reveal that the relative angle between the STAS domain and the TM domains varies, suggesting that movement of the STAS may contribute to anion transport (Supplementary Fig. 10c). Therefore, we examined interactions between the TM and STAS domains in ssPendrin.

There are notable interactions between the TM and STAS domains in ssPendrin. Firstly, at the scaffold domain/STAS interface, a density appears near α3 and close to TM14 that likely is a tightly bound ion of unknown identity in the STAS domain (Fig. 4a–c). The ion binding site is lined by residues His698 and Lys702. However, mutation H698A has no effect on either I$^-$ or HCO$_3^-$ transport, while K702E reduces HCO$_3^-$ transport by 43.7% but has no effect on I$^-$ transport (Fig. 4d, e). Also, at the scaffold domain/STAS interface, there are interactions between Gln474 and Cys706, Asn475 and Lys702, and Val670 and Gln705

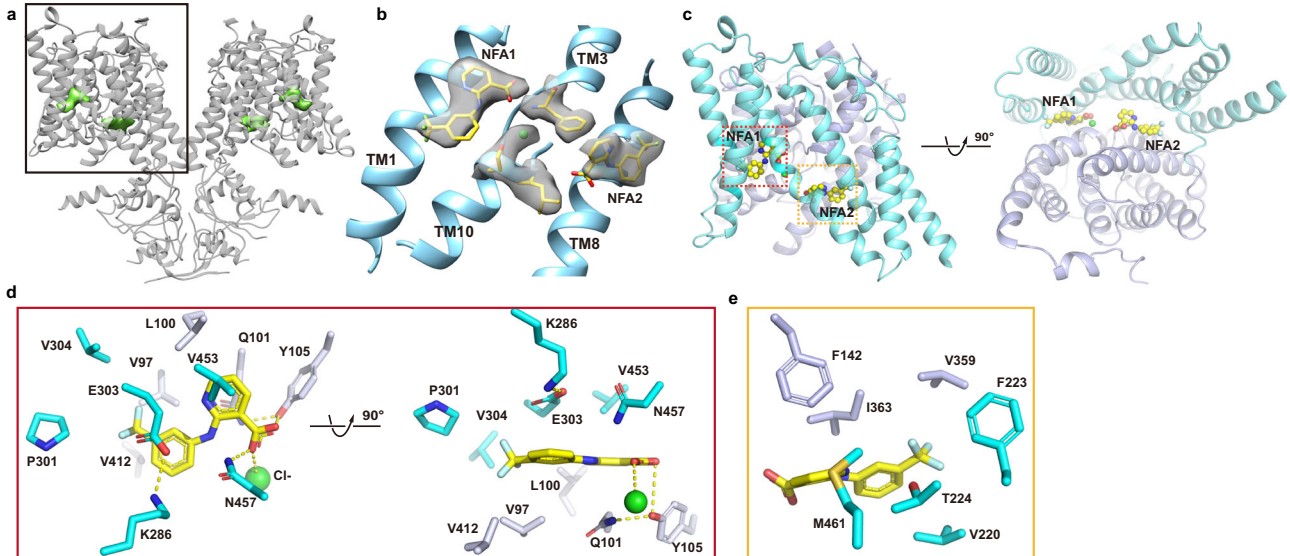

**Fig. 5 | Niflumic acid binding sites on pendrin. a** Structure of ssPendrin (cartoon) in the presence of Cl⁻ and NFA. Densities for NFA are shown as green surface. **b** View of the transport domain with densities for the two NFA, NFA1 and NFA2, and the Cl⁻ at S1. NFA is shown as sticks and Cl⁻ is shown as green sphere. **c** View of the NFA binding sites from the intracellular side. **d** NFA1 and its binding pocket. **e** NFA2 and its binding pocket.

(Fig. 4c). Mutation N475V shows a modest reduction of $HCO_3^-$ and I⁻ transport (Fig. 4d, e). Additionally, between the scaffold domain and STAS domain are interactions involving TM5, TM14 and IL1 with α1 and α2 (Fig. 4f). Notably, Tyr214 interacts with Gln514, Arg213 interacts with both Asp560 and Gly557, and Asp217 interacts with Arg674. Mutation D560V shows a moderate reduction of $HCO_3^-$ and I⁻ transport (Fig. 4d, e).

Secondly, residues 570-653 from the STAS domain, often referred to as the intervening sequence (IVS), is located between α1 and β4, and the first 15 residues (570-585) are resolved as an α-helix (α1E) while the rest is unresolved (Fig. 4g). The helix is near the intracellular loop between TM8 and 9 of the transport domain, and residues Asp376 and Thr378 on the loop could interact with residues Arg576, Asn579, and Lys580 on the IVS (Fig. 4g). Mutations to these residues significantly reduce $HCO_3^-$ or I⁻ transport, indicating that interactions between the IVS of STAS and the transport domain may affect anion transport (Fig. 4d, e). Combined, these results reinforce the notion that the STAS domain is involved in the function of pendrin. Consistent with this notion, many Pendred syndrome-associated mutations are found on the STAS domain.

## Inhibition by niflumic acid

The FDA-approved anti-inflammatory drug niflumic acid (NFA), which is a cyclooxygenase-2 inhibitor, is also one of the most effective known inhibitors of pendrin[14]. We confirmed that NFA inhibits I⁻ and $HCO_3^-$ transport in ssPendrin (Fig. 1h, j) with an IC50 of ~15 μM (Fig. 1h). We then determined the structure of ssPendrin in the presence of Cl⁻ and NFA to a resolution of 3.0 Å (Fig. 5a, Supplementary Fig. 7). While the overall structure of the NFA-bound ssPendrin is similar to the structures without the inhibitor, we found two large densities in each protomer that fit NFA well, and we defined the two sites as NFA1 and NFA2 (Fig. 5b, c). NFA1 is coordinated by Tyr105 from the transport domain and Asn457 from the scaffold domain, and the carboxylate group of NFA occupies the S2 Cl⁻ binding site (Fig. 5d). Consequently, the NFA bound ssPendrin has only one anion (S1) bound. NFA2 is less well-coordinated, with its aromatic ring wedged between the transport and scaffold domains (Fig. 5e). Since the intracellular gap between the transport and scaffold domains is expected to close in the outward-facing conformation, binding of NFA would bias the structure of pendrin in the inward-facing conformation. To further examine the

role of NFA on pendrin's function, we performed mutagenesis studies targeting NFA binding site residues (Supplementary Fig. 2c,d). Even in the absence of NFA, many mutants did not retain significant transport activity, precluding test of NFA inhibition. We measured the inhibition of transport for mutants Q101L (NFA1 site) and T224V (NFA2 site) (Supplementary Fig 2b,c), which retain measurable ¹⁴C-$HCO_3^-$ transport activity. The Q101L mutation reduces the $HCO_3^-$ or I⁻ transport activity by about 50% (Fig. 3h,i), and has an ~two-fold reduction in inhibition, shifting the IC50 from 15.5 μM in the WT to 31.2 μM for Q101L. The IC50 for T224V was 9.5 μM, and it was 16.3 μM for Q101L/T224A. The modest effect on IC50 indicates the limitation of our functional assay and further studies are required to assess and validate the presence of two NFA binding sites.

## Discussion

The structures of ssPendrin are captured in an inward-facing state, in which the substrate binding sites are solvent accessible from the intracellular side. Structural studies of SLC4 and SLC26 families of transporters have led to the consensus that the transport domain undergoes a rigid-body motion while the scaffold domain remains relatively static to expose the anion binding site to either side of the membrane, and this is commonly referred to as the elevator mechanism[41,53]. Alignment of the scaffold domain of ssPendrin to that of prestin[20] in an occluded state (Fig. 6a) show that a modest movement of the transport domain in ssPendrin, ~6.2 Å, could expose the anion binding site to the extracellular side (Fig. 6b). However, a structure of pendrin in the outward-facing state is needed to fully understand the conformational changes the transport domain undergoes.

The structures of ssPendrin led us to conclude that there are two anion binding sites in pendrin. While S1 has been consistently observed in anion transporters of the same structural fold, S2 has not been reported previously. The residues involved in S1 and S2 are highly conserved among all SLC26 members (Supplementary Fig. 12). S1 and S2 are ~5 Å apart and appear to have similar ion occupancy based on the density map, which led us to speculate that the two sites can be occupied simultaneously in the inward facing conformation. Saturation binding studies using ¹²⁵I⁻ support that pendrin under equilibrium conditions can bind two anions (e.g., I⁻) simultaneously (Fig. 3g). However, we notice that although S1 is contained within the transport

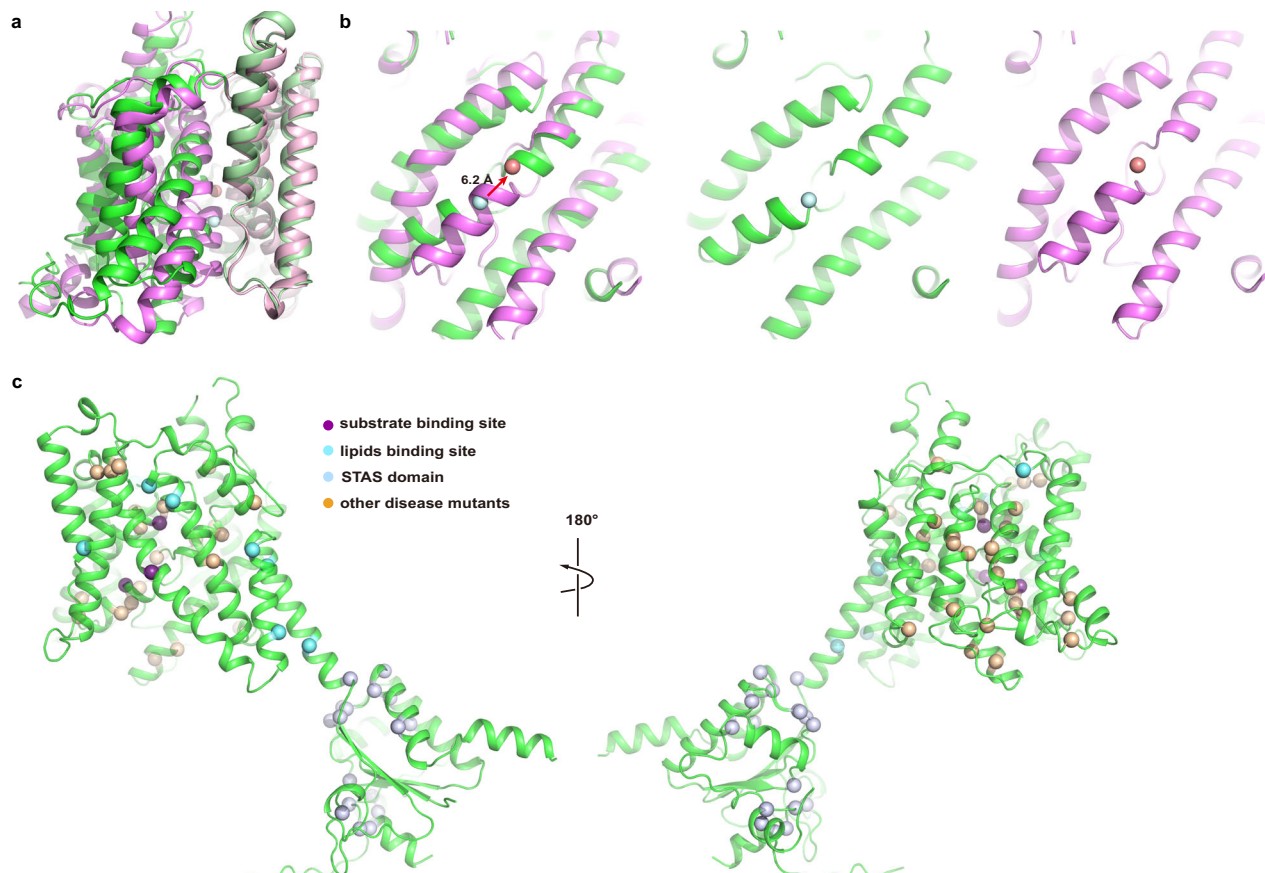

**Fig. 6 | Alignment with prestin and Pendred syndrome mutations. a** Alignment of ssPendrin and prestin (magenta) (PDB ID: 7LGU)[20] by the scaffold domains; **b** The alignment in **a** is rotated to show the relative positions of the transport domains. The Cl⁻ (green sphere) at the S1 site is shown. **c** ssPendrin (cartoon) with Pendred syndrome mutants shown as spheres.

domain, S2 involves both domains with the side chain of Asn457 from the scaffold domain contributing to its coordination. Since changing to the outward-facing conformation requires movement of the transport domain relative to the scaffold domain, S2 may be compromised in the outward-facing conformation if loss of coordination from Asn457 is not replaced. Whether the two anion binding sites are preserved in the outward-facing conformation will affect the stoichiometry of anion exchange, and further studies are required to unravel the mechanism of anion recognition, selectivity, and exchange in pendrin.

The function of the STAS domain remains a mystery. In a plant homolog of SLC26, mutations and deletion of the STAS domain reduce transport or surface expression level[26]. In SLC26A9, deletion of the IVS enhances the channel activity[18]. In the current study, we found that point mutations that disruption of interactions between the IVS and transport domain reduces transport activity (Fig. 4d, e, g). The STAS domains in the structures of various members of the SLC26A family assume different orientations relative to their TM domains (Supplementary Fig. 11), suggesting that movement of the STAS domain is involved in the process of anion transport. Additional studies are needed to uncover the role of the STAS domain.

The structure of ssPendrin in complex with NFA is captured with Pendrin in an inward-facing state and the two identified NFA molecules occupy the space between the transport and scaffold domains. We are aware that human Anion Exchanger 1 (hAE1) in complex with NFA was captured in an outward-facing state with a single NFA filling the space between the transport and scaffold domains (Supplementary Fig. 13)[37]. However, when we model the outward-facing state of Pendrin by aligning its transport and scaffold domains onto these of hAE1, we

found that the NFA binding site would be occluded by the helical EL1, which is much longer than that of hAE1 (Supplementary Fig. 13). It seems that NFA can only bind to Pendrin in the inward-facing state, and capturing Pendrin in an outward-facing conformation should provide more clarification on this issue.

We mapped the Pendred syndrome mutations onto the structure of ssPendrin (Fig. 6c). Many of these genetic mutations are in or around the anion binding pocket, likely affecting anion binding. The STAS domain also hosts many Pendred syndrome mutations, indicating its crucial role in the structure and function of pendrin. In summary, we anticipate that results from the current study will open avenues of research that will lead to an improved mechanistic understanding of anion recognition and exchange, its regulation by the STAS domain, unraveling the structural and functional impact of mutations causing Pendred syndrome, and facilitate the development of drug-based therapies that target pendrin.

## Methods
### Cloning, expression, and purification of pendrin
The *Sus scrofa* pendrin gene (NCBI accession number XM_003357511) was codon-optimized and cloned into a pFastBac Dual expression vector with a C-terminal TEV protease cleavage site (ENLYFQG) followed by 8x His-tag for production of baculovirus according to the Bac-to-Bac method (Thermo Fisher Scientific). P3 or P4 viruses were used to infect High Five (*Trichoplusia ni*) and Sf9 (*Spodoptera frugiperda)* insect cells at a density of about 3 ×10⁶ cell/mL, and the infected suspension cells were grown at 27 °C for about 64 h. Harvested cell pellets underwent a hypotonic/hypertonic wash protocol as previously described. In brief, cells were initially lysed in a hypotonic buffer

containing 10 mM 4-(2-hydroxyethyl)−1-piperazineethanesulfonic acid (HEPES), pH 7.5, 10 mM NaCl, 2 mM β-mercaptoethanol (BME), and 1 mM phenylmethylsulfonyl fluoride (PMSF), and centrifuged at 55,000 × g for 10 min at 4 °C. The pelleted cell membranes were then resuspended in a hypertonic buffer containing 20 mM HEPES, pH 7.5, 1 M NaCl, 2 mM BME, 1 mM PMSF, and 25 mg DNase I, and were centrifuged again at 55,000 × g for 20 min. Purified cell membranes were homogenized in 20 mM HEPES, pH 7.5, 150 mM NaCl, 2 mM BME, 10% (v/v) glycerol and an EDTA-free protease inhibitor cocktail tablet (Roche), and then they were flash-frozen in liquid nitrogen for storage.

Thawed purified membranes were solubilized with 1.5% (w/v) lauryl maltose neopentyl glycol (LMNG, Anatrace) at 4 °C for 2 h with moderate shaking. Detergent solubilized protein was separated from cell debris by centrifugation (55,000 × g, 45 min, 4 °C). The protein was attached to cobalt-based affinity resin (Talon, Clontech) at 4 °C for 1 h, with 10 mM imidazole added. The beads were washed with 20 mM HEPES, pH 7.5, 150 NaCl, 2 mM BME, 10% (v/v) glycerol, 0.1% (w/v) LMNG, 0.01% cholesteryl hemisuccinate (CHS), and 20 mM imidazole. The purified protein was eluted with the same buffer containing 300 mM imidazole, followed by concentration (Amicon 100 kDa cutoff, Millipore) and loading onto a size-exclusion column (SRT-3C SEC-300, Sepax Technologies) equilibrated with 20 mM HEPES, pH 7.5, 150 mM NaCl, 2 mM BME, and 0.1% (w/v) LMNG, 0.01% CHS. For the samples used in cryo-EM, the size-exclusion column was equilibrated with 20 mM HEPES, pH 7.5, 2 mM BME, 0.02% GDN (Anatrace), and 150 mM of the corresponding salt. Grids for Pendrin-Cl⁻ used 150 mM NaCl, Pendrin-I⁻ used 150 mM NaI, Pendrin-HCO₃⁻ used 150 mM NaHCO₃, Pendrin-gluconate used 150 mM NaGluconate, and Pendrin-NFA used 150 mM NaCl and 10 mM NFA.

Pendrin mutants were generated using the QuikChange method (Stratagene) and the cDNA was sequenced to verify the mutation. Expression and purification of mutants followed the same protocol as for ssPendrin-WT.

## Cryo-EM sample preparation and data collection
Cryo grids were prepared on the Thermo Fisher Vitrobot Mark IV. Quantifoil R1.2/1.3 Cu grids were glow-discharged using the Pelco Easyglow. Concentrated ssPendrin protein in different buffers (3.5 µL) was applied to glow-discharged grids. After blotting with filter paper (Ted Pella) for 4.0-4.5 s, the grids were plunged into liquid ethane cooled with liquid nitrogen. For cryo-EM data collection, movie stacks were collected on a Titan Krios at 300 kV with a Quantum energy filter (Gatan), at a nominal magnification of ×81,000 or ×105000 and with defocus values of -2.0 to -0.8 µm. A K3 Summit direct electron detector (Gatan) was paired with the microscope. Each stack was collected in the super-resolution mode with an exposing time of 0.175 s per frame for a total of 50 frames. The dose was about 50 e⁻ per Å² for each stack.

## Cryo-EM data processing
The stacks were motion-corrected with Relion and binned (2 × 2)[54]. Dose weighting was performed during motion correction, and the defocus values were estimated with Gctf[55].

For Pendrin-Cl⁻/I⁻/HCO₃⁻/gluconate/NFA data set, a total of 7,390,265/5,134,368/3,676,424/8,803,318/8,806,745 particles were automatically picked in RELION 3.1[56] with template picking from 4339/4920/3870/8828/5106 images and imported into cryoSPARC[57]. After three rounds of 2D classification, 35/20/25/25/32 classes (containing 1,407,280/472,656/425,277/969,097/592,263 particles) were selected out of 200 2D classes for ab initio three-dimensional 3D reconstruction, which produced one good class with recognizable structural features and three bad classes that did not have structural features. Both the good and bad classes were used as references in three rounds of the heterogeneous refinement (cryoSPARC) and yielded a good class at 3.39/3.9/3.5/3.4/3.8 Å from 760,950/163,394/202,035/494,226/437,899 particles. Then nonuniform refinement (cryoSPARC) was

performed with C2 symmetry and an adaptive solvent mask, CTF refinement yielded a map with an overall resolution of 2.5/2.8/2.7/2.5/3.0 Å. Resolutions were estimated using the gold-standard Fourier shell correlation with a 0.143 cut-off[58] and high-resolution noise substitution[59]. Local resolution was estimated using ResMap[60].

## Model building and refinement
The structural model of ssPendrin in complex with Cl⁻ were built de novo into the density map starting with poly-alanine, and sidechains were then added onto the model based on the map. The structures models of ssPendrin in complex with I⁻/HCO₃⁻/gluconate/NFA was built based on the model of ssPendrin-Cl⁻. Model building was conducted in Coot[61]. Structural refinements were carried out in PHENIX in real space with secondary structure and geometry restraints[62]. The EMRinger Score was calculated as described[63].

## Scintillation proximity assay (SPA)-based binding
Purified ssPendrin variants were bound to Ni²⁺ chelate scintillation NanoSPA beads (Scintillation Nano Technologies, Inc) following established protocols[46]. Briefly, 100 µg NanoSPA beads were used per assay in a volume of 100 µL assay buffer composed of 500 mM Tris/Mes, pH 7.5, 5% glycerol, 0.1 mM TCEP, 0.1% DDM, 0.006% CHS with 150 ng purified protein. Binding of ¹²⁵I⁻ at the indicated concentrations (0.1 Ci/mmol; American Radiolabeled Chemicals, Inc.) was assayed for 1 h at 23 °C before measuring the samples in a MicroBeta photomultiplier tube microcounter (Wallac). Nonspecific binding was assayed in the presence of 800 mM imidazole. Specific binding was determined by subtracting the nonspecific binding from the total binding and was plotted as a function of free radioligand. Nonlinear regression fitting of the data was performed in Prism 8 to obtain the $EC_{50}$, the Hill coefficient, and the molar ratios using the 'Specific binding with Hill slope' model.

## Proteoliposome preparation
POPE and POPG lipids dissolved in chloroform (Avanti) were mixed at a 3:1 (w/w) ratio and with 10 mol% cholesterol. Lipids were dried under a stream of argon gas, with trace chloroform removed under vacuum for two hours. Dried lipids were rehydrated with 20 mM HEPES, pH 7.5 and 150 mM NaCl to a final concentration of 10 mg lipids per ml buffer. Rehydrated lipids were sonicated until transparent, and 40 mM n-decyl-β-D-maltopyranoside (DM) was then added. WT or mutant ssPendrin variants were added at a 1:100 (w/w, protein:lipid) ratio, incubating at 4 °C while shaking for 15 minutes. Detergent was removed by dialysis at 4 °C for four days, changing 1 L buffer each day[64]. After detergent removal, the liposomes were aliquoted and flash-frozen in liquid nitrogen for further use.

## ¹⁴C-HCO₃⁻ and ¹²⁵I- uptake assays
Before the uptake experiments, proteoliposomes or control liposome aliquots were subjected to three freeze/thaw cycles followed by extrusion through a 400 nm filter membrane (NanoSizerTM Extruder, T&T Scientific Corporation) to obtain a homogeneous suspension. (Proteo)liposomes were diluted and the outside buffer was exchanged through a PD−10 desalting column (Cytiva) into 20 mM HEPES, pH 7.5, 150 mM sodium gluconate. Experiments were performed at 37 °C and uptake reactions were initiated by the addition of 100 µM ¹⁴C-HCO₃⁻ (in the form of NaHCO₃) or 10 µM ¹²⁵I⁻ as NaI; both radiolabeled compounds were purchased from American Radiolabeled Chemicals, Inc. and used as specific activities as appropriate. Reactions were stopped by the addition of quenching buffer (at 23 °C) containing 20 mM HEPES, pH 7.5 and 150 mM NaHCO₃ to prevent backflow (for uptake experiments involving ¹⁴C-HCO₃⁻) or ice-cold 20 mM HEPES, pH 7.5, 150 mM sodium gluconate (for ¹²⁵I⁻ uptake experiments) and were filtered through 0.45 µm nitrocellulose filters (Millipore). The radioactivity retained on the filters was determined by liquid scintillation

counting. A standard curve was plotted with known amounts of $^{14}C$-$HCO_3^-$ or $^{125}I^-$ to convert counts per minute to pmol of $HCO_3^-$ or $I^-$. Each data point represents the mean ± s.e.m. values from at least three repeats. Experiments with different $Cl^-$ gradients were prepared with (proteo)liposomes rehydrated in 20 mM HEPES, pH 7.5, 150 mM sodium gluconate. For experiments with inhibitors, NFA at the indicated concentrations was added to proteoliposomes prior to the addition of $NaHCO_3$, incubating for 5 minutes at 37 °C.

The kinetics of $^{125}I^-$ uptake by ssPendrin-WT were determined by measuring the initial rates of transport for 10-s periods (the 10-s signal was corrected for the background signal measured at 0 s) at varying $^{125}I^-$ concentrations (1 Ci/mol). The specific uptake data were normalized to the actual protein content of the proteoliposomes using routine protein determinations[65]. Data were plotted as function of $I^-$ concentration in Prism 8 and fit to the Michaelis-Menten equation, yielding the $K_m$ and maximum velocity of transport ($V_{max}$ or $k_{cat}$).

### Fluorescence-based iodide uptake assay
Before the uptake experiments, liposome aliquots were diluted to also contain 0.125 mM calcein and went through three rounds of freezing and thawing and were extruded to homogeneity. Liposomes were diluted and the outside buffer was exchanged through a PD-10 desalting column into 20 mM HEPES, pH 7.5, 150 mM sodium gluconate. Liposomes were incubated at 37 °C for 5 minutes prior to initiation of transport by addition of 10 mM NaI. For experiments with the inhibitor, NFA was added during this incubation. Fluorescence intensity was measured at emission 513 nm with excitation 494 nm at 10-s intervals over 10 minutes, using the Fluoromax-4 Spectrophotometer (Horiba). Similar to $HCO_3^-$ uptake experiments, for using different $Cl^-$ gradients, liposomes were rehydrated in 20 mM HEPES, pH 7.5 and 150 mM sodium gluconate.

### Reporting summary
Further information on research design is available in the Nature Portfolio Reporting Summary linked to this article.

## Data availability
The data that support this study are available from the corresponding authors upon request. The atomic coordinates have been deposited in the Protein Data Bank (PDB) under the accession code 8SGW (ssPendrin in complex with $Cl^-$), 8SH3 (ssPendrin in complex with $I^-$), 8SIE (ssPendrin in complex with $HCO^-$), 8UUK (ssPendrin apo-state), and 8SHC (ssPendrin in complex with NFA). The electron microscopy maps have been deposited in the Electron Microscopy Data Bank EMDB under the accession codes EMD-40470 (ssPendrin in complex with $Cl^-$), EMD-40479 (ssPendrin in complex with $I^-$), EMD-40507 (ssPendrin in complex with $HCO^-$), EMD-42588 (ssPendrin apo-state) and EMD-40483 (ssPendrin in complex with NFA). The source data underlying Figs. 1b-1h, 1j-1k, 3g-3i, 4d-4e, and Supplementary Figs. 1, 2a-d are provided in the Source Data File. Source data are provided with this paper.

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

## Acknowledgements

This work was supported by grants from NIH (DK122784 to M.Z., GM145416 to A.L. and M.Z., and GM151548 to M.Q. and M.Z.). We acknowledge the cryo-EM cores in Baylor College of Medicine (CPRIT Core Facility Award RP190602) for their support in grid preparation and screening. We are grateful to the Pacific Northwest Center for Cryo-EM supported by NIH grant U24GM129547, the National Center for CryoEM Access and Training (NCCAT), and the Simons Electron Microscopy Center supported by the NIH grant U24GM129539 and by grants from the Simons Foundation (SF349247), Stanford-SLAC Cryo-EM Center (S2C2) supported by the NIH grant U24GM129541, Laboratory for Bio-Molecular Structure (LBMS) supported by the DOE Office of Biological and Environmental Research (KP1607011), and Laboratory for Biomolecular Structure and Dynamics (LBSD) of Texas A&M University for the support in data collection.

## Author contributions

M.Z. and M.Q. conceived the project. L.W., A.H., E.G., A.L., M.Q., and M.Z. designed the experiments. L.W., A.H., and E.G. conducted experiments. L.W., A.H., E.G., A.L., M.Q., and M.Z. analyzed the data and wrote the manuscript.

## Competing interests

The authors declare no competing interests.
