## [Peer Review File · Nature Communications]

Mechanism of anion exchange and small-molecule inhibition of pendrinReviewers' Comments:

Reviewer #1:

Remarks to the Author:

The authors report the Cryo EM structures of Pendrin (SLC26A4), which is an anion exchanger that mediates the transport of bicarbonate ions (HCO_3^-) in exchange for chloride ions (Cl^-) and has important physiological roles. The Cryo EM structures indicate two anion binding sites, which is partially supported by transport activities in liposomes and mutagenesis. The authors also report the structure of SLC26A4 in complex with an anti-inflammatory drug niflumic acid (NFA). Overall, this is an important contribution with most conclusions partially supported by functional data. Certainly further validation of the S2 site and the functional impact for a potential change in substrate stoichiometry is required- otherwise the mechanistic insights still seem a bit limited here overall; despite the superb work to get to this stage.

-Transport data-

The controls show little ^{14}C - HCO_3^- uptake in either empty liposomes or liposomes without internal Cl^- . Its very nice data, however, taking 10 mins to transport ~ 250 molecules seems a bit slow. Has rates been adjusted for protein reconstitution efficiency? It was interesting to read you reconstitution protocol into liposomes and I haven't seen an reconstitution using long dialysis into DM-pre-solubilised liposomes used before nor it is typical to freeze-thaw proteoliposomes several times, although I understand its good if this works as you can get tighter proteoliposomes.

What is the K_m for HCO_3^- ? Is the activity of pendrin pH dependent?

- Structure and S2 site-

S2 binding site is new and needs further validation. The obvious critique is that it might be a water molecule. How many "other" densities could be modelled as water in the maps? How is the ConSurf analysis of the residues coordination the S2 anion? I would suggest using the CheckMyMetal website, which I think would provide some further confidence in identifying the most likely ion and its coordination (<https://cmm.minorlab.org>). The mutation of R409L seems a bit more drastic than an R409H or even a R409A mutation. It was unclear to me why hydrophobic substitutions have been made in most cases rather than alanine? The Y105F mutation didnt abolish I^- transport, which might be expected for a critical coordinating residue. How about an Y105A mutation? Given the challenges with validation of this ion binding site it might be the most straightforward to determine a Cryo EM structure of a mutation to one of the S2 coordinating residues to show loss of additional map density. The authors write that "further studies are required to unravel the impact of two anion binding sites on the mechanism of anion recognition, selectivity, and exchange in pendrin". I understand that ion coupling will take time to fully piece together, but more data is needed to support the additional site. It would helpful if the authors could clarify how 2 anion sites would work in a functional context, i.e., pendrin is thought to be an electroneutral exchanger and so what is the rationale for $2\text{HCO}_3^- : 2\text{Cl}^-$ exchanger? In your transport assays is there any difference in transport upon the addition of valinomycin?

-Lipids-

There are additional lipids other than cholesterol model. It wasnt clear to me that the map density was clear enough to assign these different lipids? Cholesterol is thought to be predominantly in the outer leaflet and this is consistent with the recent structure of an ion channel in native vesicles from the MacKinnon lab (<https://www.pnas.org/doi/10.1073/pnas.2302325120>). For this reason, cholesterol sites in the inner leaflet need extra scrutiny. In this case, CHS was added to the purification and also GDN in the last step, which has a cholesterol backbone. The cholesterol next to scaffold domain on the inner leaflet may not be physiologically relevant. In this case, an A270W mutation showed no sensitivity to CHS addition, but the mutation also showed no transport activity. Its possible that this the A270W has perturbed transport by pendrin that is not related to cholesterol sensitivity. The functional data supports CHS addition increases activity of pendrin in liposomes, but this is also

somewhat to be expected since the plasma membrane has 50:50 mol fraction of cholesterol.

-STAS domain-

The two mutations showing the most obvious loss in transport activity are on the transport domain (N475V and D560V) also had the poorest SEC traces, with breakdown into what appears to be monomers. Was the reconstitution efficiency of the mutants into liposomes also compared? Is there a reason why the point mutations seem to have a stronger impact on HCO₃ transport vs I⁻ transport? Overall, this is a solid section, but as highlighted by the authors, it doesn't reveal the functional reason for the STAS domain. Indeed, I find it somewhat surprising that the K702E mutation doesn't seem to have an impact on transport activity.

-inhibitor bound structure-

Was it expected that there would be two binding sites for NFA? Mutagenesis would be helpful to clarify if both sites have a functional impact. Is Cl⁻ binding required for NFA inhibition?

Reviewer #2:

Remarks to the Author:

The manuscript by Wang et al. reports the first structures for Pendrin, an SLC26 family transporter responsible for exchanging chloride and bicarbonate in kidney, lung and cochlea cells. It is well established that the counter transport of anions is essential in regulating intracellular pH, and indeed, mutations in Pendrin lead to several physiological conditions linked to acid/base dysregulation. Specifically, the study reports the cryo-EM structures of Pendrin in the inward (cytoplasmic facing conformation) bound to Cl⁻, I⁻ and HCO₃⁻ anions. In addition, the authors also present the structure of Pendrin bound to Niflumic acid (NFA), which has been used in the clinic to inhibit cyclooxygenase-2 but has also been shown to have off-target effects by inhibiting anion channels, including Pendrin. The structural work is supported by several in vitro functional assays, which are used to validate the structures, support the proposed mechanisms and inform on the role of cholesterol as a likely component of the fully functional transporter. Overall the work is of a very high standard, and the manuscript is well-written and clear to follow. In addition to the novelty with respect to the Pendrin structure, the work also reveals a novel second anion binding site, S2, which has not been observed in other structurally similar members of the SLC26 family (such as Prestin) or SLC4 bicarbonate exchanges. In addition to the insights gained by mapping the disease-causing mutations associated with Pendrin, and the mechanism of inhibition by NFA, the work is very suitable for Nature Communications.

I am very much in favour of publishing this paper, and my comments are there to assist the authors in considering areas where additional clarification might help the uninitiated in solute carrier research. My main comment concerns:

1. Lines 123- 153 - The functional validation of the S2 anion binding site. This observation seems one of, if not the most significant, discoveries made by the authors in this report. Yet, the functional validation of this site is somewhat brief. I appreciate the data on the various binding site mutants in Fig. 3f and 3g; however, given the extensive effort to establish several functional assays detailed in Fig. 1, it seemed like a missed opportunity to apply some of these to dissecting the importance of S2 to the Pendrin mechanism. For example, what impact does Q101L have on Pendrin? Does it reduce the turnover number or the K_m for anions? In line 74, the authors demonstrate that their radioactive assay can measure the number of HCO₃ molecules moved per Pendrin in 10 minutes. So what is the number for the S2 site mutants? Incidentally, do the authors know that every Pendrin is equally active in the assay? Maybe some Pendrins are super active, and others are inactive. Given the requirement

for cholesterol and an interesting relationship with lipids, I think this might be a confounding factor. Nevertheless, such an assay on selective mutants in the S2 site would still be informative.

2. Similarly, Lines 80-81 – The authors present data in Fig. 1d concerning the stimulatory effect of an outwardly directed Cl⁻ gradient on HCO₃⁻ uptake. However, very little detail is given on this observation's implications for Pendrin's physiological function. In addition, if Cl⁻ can drive HCO₃⁻ uptake, the two anions must be coupled. In short, I was confused about what the authors were trying to tell me about the mechanism of anion transport by Pendrin with this data. I would suggest expanding the explanation or the data to show what impact the Cl⁻ gradient has. It should be straightforward to show that HCO₃⁻ can drive uptake of I⁻ against an opposing I⁻ concentration gradient using the fluorescent assay shown in Fig. 1h Or the ¹⁴C labelled HCO₃⁻. I don't think this is an essential set of experiments for the thrust of the study, but as it stands, the authors raise the question, present some data which doesn't definitively answer the question of coupling and then leave the question hanging.

Minor comments:

Line 92 – How did the authors assign the density for Cl⁻? Was this via comparison to the other SLC26 family members or via the hydrogen bond length? I would also consider labelling the helices in Fig. 3a-c to facilitate orientation by the reader.

Line 107 – please reference at least one of the 'previous publications'.

Line 117 – missing 'for'.

Line 121-22. This sentence seems unnecessary. Would anyone have any reason to call into question the topology of Pendrin, given the sequence and structural similarity to other SLCs?

Line 136 – Can the interaction between F141 and the anion in site S1 strictly be called an anion-π? I thought this term was for interactions with the delocalised electron cloud on the face of aromatics.

Lines 139-144 – this section seems out of context. I suggest rewriting it to make it more straightforward for the reader. For example, the fact about R409H appears bolted on. Incidentally, in Pendred syndrome, is Pendrin correctly trafficked but non-functional, or do these mutations cause folding defects? Similarly, for Lines 147-149.

Lines 163 – Do the mutations around the unknown ion binding site impact protein stability or expression? This data might inform on a structural function for the novel ion in the STAS domain. It would also be interesting to compare this region with the other STAS-containing structures. Is this feature unique to Pendrin?

Line 179 – I wondered whether there was any correlation between the severity of Pendrin syndrome and the location of the mutations. Indeed, are there levels of severity in the disease?

Line 207 – the authors suggest the mechanism of NFA inhibition is simply jamming the transporter in the inward open state. However, I'm left wondering whether NFA can bind the outward open state of the transporter and then clog in the inward-facing state or whether NFA enters the cell via another route and then jams Pendred. Could the authors please elaborate on how they think NFA enters the cell and interacts with Pendred? In addition, given that homologous structures of SLC4 and SLC26 structures have been determined in both outward and inward-facing states, do the authors think a structural comparison showing how NFA would hypothetically jam Pendred using a homology model of the outward-facing state in the SI (linked to the structural comparison in Fig. 7) might illustrate this point more clearly?

REVIEWER COMMENTS

Reviewer #1 (Remarks to the Author):

The authors report the Cryo EM structures of Pendrin (SLC26A4), which is an anion exchanger that mediates the transport of bicarbonate ions (HCO_3^-) in exchange for chloride ions (Cl^-) and has important physiological roles. The Cryo EM structures indicate two anion binding sites, which is partially supported by transport activities in liposomes and mutagenesis. The authors also report the structure of SLC26A4 in complex with an anti-inflammatory drug niflumic acid (NFA). Overall, this is an important contribution with most conclusions partially supported by functional data. Certainly further validation of the S2 site and the functional impact for a potential change in substrate stoichiometry is required- otherwise the mechanistic insights still seem a bit limited here overall; despite the superb work to get to this stage.

We thank the Reviewer for the overall positive evaluation of our manuscript.

Comment 1:

The controls show little ^{14}C - HCO_3^- uptake in either empty liposomes or liposomes without internal Cl^- . Its very nice data, however, taking 10 mins to transport ~ 250 molecules seems a bit slow. Has rates been adjusted for protein reconstitution efficiency? It was interesting to read you reconstitution protocol into liposomes and I haven't seen an reconstitution using long dialysis into DM-pre-solubilised liposomes used before nor it is typical to freeze-thaw proteoliposomes several times, although I understand its good if this works as you can get tighter

proteoliposomes.

We thank the Reviewer for the critical comments and like to address the comments in the following manner: a) reconstitution protocol – we used DM-pre-solubilised liposomes to avoid potential negative impact by more commonly used shorter chain detergents such as β -octylglucoside. We found that extensive dialysis clears DM and the vesicles are tight. This protocol otherwise is identical in principle to a fairly well-established one that was previously described¹. In parallel, we also performed reconstitution using detergent-destabilized pre-formed liposomes followed by a BioBead-based detergent extraction that has been successfully used in the Quick lab for many applications. Both preparations yield comparable results. Several freeze-thaw cycles for proteoliposomes followed by sonication and/or extrusion have yielded consistently the best results in numerous applications. b) transport rates – we agree with the slow uptake activity; however, the working concentration of $^{14}\text{C-HCO}_3^-$ is far below the K_m of transport, so the shown figure does not reflect the maximum transport activity. The specific activity of commercially available $^{14}\text{C-HCO}_3^-$ precludes performing concentration-dependent transport measurements that reach saturation while simultaneously providing a satisfactory signal-to-noise ratio to reliably determine the V_{max} . As an alternative, we took advantage of the higher energy of $^{125}\text{I}^-$, another substrate of pendrin, to measure such concentration-dependent uptake experiments with favorable signal-to-noise ratio even at high mM concentrations. We have included a new figure to show these measurements (Fig. 1g). The K_m of I^- transport is in the 10 mM-range; transforming the V_{max} of the uptake reaction into the catalytical turnover number (k_{cat}) using accurate protein determinations in the proteoliposome preparations to transform μg into pmol as we routinely use (as previously described in ref²) yielded a k_{cat} of about 1.2 s^{-1} . Such a k_{cat} is common for secondary transporters; several other transporters

with a k_{cat} in the sub 1 s^{-1} range were reported³. c) effect of anions – see response to {Reviewer#2 Comment#2}

Comment 2:

What is the K_m for HCO_3^- ? Is the activity of pendrin pH dependent?

As stated above, the specific activity of the commercially available $^{14}\text{C-HCO}_3^-$ precluded these measurements, and we have used $^{125}\text{I}^-$ instead to present information about pendrin's transport affinity. To correlate the kinetics of pendrin-mediated I^- transport with the protein's binding activity, we have performed binding studies using $^{125}\text{I}^-$ as well.

Our preliminary data, supporting previously reported experimental evidence, indicate that OH^- is a substrate of pendrin^{4,5}. Given the current status of these studies, we prefer to defer to a follow up manuscript to report extensively on the role of pH (OH^-) on pendrin's function.

Comment 3:

S2 binding site is new and needs further validation. The obvious critique is that it might be a water molecule. How many "other" densities could be modelled as water in the maps? How is the ConSurf analysis of the residues coordination the S2 anion? I would suggest using the CheckMyMetal website, which I think would provide some further confidence in identifying the most likely ion and its coordination (<https://cmm.minorlab.org>).

We agree with the Reviewer on the need to further validate our claim of the existence of the S2 anion site. We have tackled this three-fold: a) with saturation binding studies using pendrin-WT (with intact S1 and S2 sites), pendrin-S408A (only intact S2 site), and pendrin-Q101L (only intact S1 site). As with the transport measurements, we took advantage of the well-suited high-energy isotope $^{125}\text{I}^-$ to perform scintillation proximity-based saturation binding studies that have been established in the Quick lab, following a procedure that was first published in 2008 (ref⁶). The results of this approach show that pendrin-WT exhibits a binding stoichiometry of 2 mol I⁻ per mol of pendrin, whereas each variant with an impaired ion binding site feature a molecular I⁻-to-pendrin binding stoichiometry of 1. The mutations were selected to target side chains in each site that interact specifically with the anion in each site, and the data of these studies indicate that one molecule of pendrin with only one intact ion binding site can still bind one I⁻. b) solving the structure of pendrin in apo conformation by preparing the protein in the presence of the ‘inert’ anion gluconate. The loss of density in the S2 site in the apo (anion-free) structure supports our interpretation that the densities in the S1 and S2 sites can be assigned to two anions. c) We also performed ConSurf analysis of S2 binding site residues and observe that Gln101 and Asn457 are highly conserved in SLC26, while Tyr105 is less conserved, sometimes appearing as Phe. The CheckMyMetal site seems more specialized in cations and does not yield good results on anions.

Comment 4:

The mutation of R409L seems a bit more drastic than an R409H or even a R409A mutation. It was unclear to me why hydrophobic substitutions have been made in most cases rather than alanine? The Y105F mutation didnt abolish I⁻ transport, which might be expected for a critical

coordinating residue. How about an Y105A mutation? Given the challenges with validation of this ion binding site it might be the most straightforward to determine a Cryo EM structure of a mutation to one of the S2 coordinating residues to show loss of additional map density.

We made the R409L mutation to neutralize the charge and preserve as much as possible the bulk of the sidechain, and the Y105F mutation to see the effect of the hydroxyl group. We agree that as much as one could rationalize the design of a mutation, structures would likely be more informative. Although we have yet to determine a structure of binding site mutations, we determined the structure of an “apo” ssPendrin by using gluconate as the only anion. The apo-Pendrin does not have density in either the S1 or S2 sites.

Comment 5:

The authors write that “further studies are required to unravel the impact of two anion binding sites on the mechanism of anion recognition, selectivity, and exchange in pendrin”. I understand that ion coupling will take time to fully piece together, but more data is needed to support the additional site. It would be helpful if the authors could clarify how 2 anion sites would work in a functional context, i.e., pendrin is thought to be an electroneutral exchanger and so what is the rationale for 2HCO₃⁻:2Cl⁻ exchanger? In your transport assays is there any difference in transport upon the addition of valinomycin?

We provided additional experimental evidence for the existence of two anion sites in pendrin. We defer to follow-up studies in which we will address the electrogenicity of pendrin’s antiport process, and this is in part due to preliminary findings on OH⁻ being a substrate. To provide more

information on the role of ‘traditional’ anions in the antiport reaction, we have performed experiments in which we tested different anions as ‘drivers’ for $^{14}\text{C-HCO}_3^-$ uptake. These studies are shown in the revised manuscript (Fig. 1f, Supplementary Fig. 2b).

Comment 6:

There are additional lipids other than cholesterol model. It wasn't clear to me that the map density was clear enough to assign these different lipids? Cholesterol is thought to be predominantly in the outer leaflet and this is consistent with the recent structure of an ion channel in native vesicles from the MacKinnon lab (<https://www.pnas.org/doi/10.1073/pnas.2302325120>). For this reason, cholesterol sites in the inner leaflet need extra scrutiny. In this case, CHS was added to the purification and also GDN in the last step, which has a cholesterol backbone. The cholesterol next to scaffold domain on the inner leaflet may not be physiologically relevant. In this case, an A270W mutation showed no sensitivity to CHS addition, but the mutation also showed no transport activity. Its possible that this the A270W has perturbed transport by pendrin that is not related to cholesterol sensitivity. The functional data supports CHS addition increases activity of pendrin in liposomes, but this is also somewhat to be expected since the plasma membrane has 50:50 mol fraction of cholesterol.

We agree with the Reviewer on this assessment. To fully address these valid concerns, we believe a more thorough analysis is needed. However, such a study would delay the revision of the manuscript and the timely publication of other findings of this work. Thus, we described the presence of lipid-like densities in the structure but refrained from assigning cholesterol to the densities. We also removed the data on A270W.

-STAS domain-

Comment 7:

The two mutations showing the most obvious loss in transport activity are on the transport domain (N475V and D560V) also had the poorest SEC traces, with breakdown into what appears to be monomers. Was the reconstitution efficiency of the mutants into liposomes also compared? Is there a reason why the point mutations seem to have a stronger impact on HC03 transport vs I-transport? Overall, this is a solid section, but as highlighted by the authors, it doesn't reveal the functional reason for the STAS domain. Indeed, I find it somewhat surprising that the K702E mutation doesn't seem to have an impact on transport activity.

The reconstitution efficiency of the mutants N475V and D560V in proteoliposomes was similar to that of the WT. The stronger effect on HCO_3^- might be due to its lower working concentration. Overall, the impact of mutations on anion transport follow a similar qualitative trends.

Comment 8:

Was it expected that there would be two binding sites for NFA? Mutagenesis would be helpful to clarify if both sites have a functional impact. Is Cl^- binding required for NFA inhibition?

This is an excellent question. We did not expect to see two NFA molecules. To further characterize the role of NFA on pendrin's function, we – as suggested by the Reviewer - performed additional mutagenesis studies, targeting NFA binding site residues (Supplementary Fig. 2c,d). Even in the

absence of NFA, many mutants did not retain significant transport activity, likely due to being so close to the binding pocket. From the mutants that retained measurable $^{14}\text{C-HCO}_3^-$ transport activity we measured the inhibition of transport for mutants Q101L (NFA1 site) and T224V (NFA2 site) (Supplementary Fig 2c,d). The Q101L mutation reduces the HCO_3^- or I^- transport activity by about 50 % (see Figure 3h,i), and has an \sim two-fold reduction in inhibition, shifting the IC_{50} from 15.5 μM in the WT to 31.2 μM for Q101L. The IC_{50} for T224V was 9.5 μM , and it was 16.3 μM for Q101L/T224A. Based on the structure, the bound Cl^- does not participate in direct coordination of NFA.

Reviewer #2 (Remarks to the Author):

The manuscript by Wang et al. reports the first structures for Pendrin, an SLC26 family transporter responsible for exchanging chloride and bicarbonate in kidney, lung and cochlea cells. It is well established that the counter transport of anions is essential in regulating intracellular pH, and indeed, mutations in Pendrin lead to several physiological conditions linked to acid/base dysregulation. Specifically, the study reports the cryo-EM structures of Pendrin in the inward (cytoplasmic facing conformation) bound to Cl^- , I^- and HCO_3^- anions. In addition, the authors also present the structure of Pendrin bound to Niflumic acid (NFA), which has been used in the clinic to inhibit cyclooxygenase-2 but has also been shown to have off-target effects by inhibiting anion channels, including Pendrin. The structural work is supported by several in vitro functional assays, which are used to validate the structures, support the proposed mechanisms and inform on the role of cholesterol as a likely component of the fully functional transporter. Overall the work is of a very high standard, and the manuscript is well-written and

clear to follow. In addition to the novelty with respect to the Pendrin structure, the work also reveals a novel second anion binding site, S2, which has not been observed in other structurally similar members of the SLC26 family (such as Prestin) or SLC4 bicarbonate exchanges. In addition to the insights gained by mapping the disease-causing mutations associated with Pendrin, and the mechanism of inhibition by NFA, the work is very suitable for Nature Communications.

I am very much in favour of publishing this paper, and my comments are there to assist the authors in considering areas where additional clarification might help the uninitiated in solute carrier research.

We appreciate not only the positive evaluation of our work by the Reviewer but also the Reviewer's constructive approach to improve the quality of the revised manuscript.

My main comment concerns:

Comment 1. Lines 123- 153 - The functional validation of the S2 anion binding site. This observation seems one of, if not the most significant, discoveries made by the authors in this report. Yet, the functional validation of this site is somewhat brief. I appreciate the data on the various binding site mutants in Fig. 3f and 3g; however, given the extensive effort to establish several functional assays detailed in Fig. 1, it seemed like a missed opportunity to apply some of these to dissecting the importance of S2 to the Pendrin mechanism. For example, what impact does Q101L have on Pendrin? Does it reduce the turnover number or the K_m for anions? In line

74, the authors demonstrate that their radioactive assay can measure the number of HCO₃ molecules moved per Pendrin in 10 minutes. So what is the number for the S2 site mutants? Incidentally, do the authors know that every Pendrin is equally active in the assay? Maybe some Pendrins are super active, and others are inactive. Given the requirement for cholesterol and an interesting relationship with lipids, I think this might be a confounding factor. Nevertheless, such an assay on selective mutants in the S2 site would still be informative.

The Reviewer's comments mirrors that of Reviewer 1's, and we think that our results on ¹²⁵I binding provide a validation for the two sites.

As for other questions in this comment, we speculate that lower transport activities observed in the binding site mutations are likely due to changes in both V_{max} and K_m . It is possible that activities of Pendrin in a vesicle are not homogeneous, and since we keep the protein lipid ratio the same for wild type and mutations, the ratio or the difference between the various protein variants are reliable.

Comment 2. Lines 80-81 – The authors present data in Fig. 1d concerning the stimulatory effect of an outwardly directed Cl⁻ gradient on HCO₃⁻ uptake. However, very little detail is given on this observation's implications for Pendrin's physiological function. In addition, if Cl⁻ can drive HCO₃⁻ uptake, the two anions must be coupled. In short, I was confused about what the authors were trying to tell me about the mechanism of anion transport by Pendrin with this data. I would suggest expanding the explanation or the data to show what impact the Cl⁻ gradient has. It should be straightforward to show that HCO₃⁻ can drive uptake of I⁻ against an opposing I⁻ concentration gradient using the fluorescent assay shown in Fig. 1h Or the ¹⁴C labelled HCO₃⁻.

I don't think this is an essential set of experiments for the thrust of the study, but as it stands, the authors raise the question, present some data which doesn't definitively answer the question of coupling and then leave the question hanging.

Those are great suggestions. The main point we wanted to make was that an opposite Cl^- gradient acts as the driving force for HCO_3^- or I^- uptake in our assays. We measured uptake of $100 \mu\text{M } ^{14}\text{C-HCO}_3^-$ into ssPendrin proteoliposomes with $1 \text{ mM } \text{Cl}^-$ outside and varying concentrations of Cl^- inside (Supplementary Fig. 2a). The steady-state amount of uptake after 10 minutes is higher with a 100-fold $[\text{Cl}^-]$ gradient compared to 50-fold. This further shows that a greater $[\text{Cl}^-]$ gradient can facilitate greater exchange of HCO_3^- . Following the Reviewer's excellent suggestion to assess how effectively other anions can facilitate exchange of HCO_3^- our substrate, we measured the uptake of $100 \mu\text{M } ^{14}\text{C-HCO}_3^-$ into ssPendrin proteoliposomes with 1 mM of the exchanged anion outside and 100 mM of the exchange anion inside to generate a 100-fold concentration gradient (Supplementary Fig. 2b). For comparison, we evaluated the initial rate of uptake (Fig. 1f). The divalent anion SO_4^{2-} is unable to drive transport while many monovalent anions are capable to driving $^{14}\text{C-HCO}_3^-$ uptake. However, there does not appear to be a clear correlation between the size of each anion and the rate of transport.

Minor comments:

Line 92 – How did the authors assign the density for Cl^- ? Was this via comparison to the other SLC26 family members or via the hydrogen bond length? I would also consider labelling the

helices in Fig. 3a-c to facilitate orientation by the reader.

The density at the S1 site was assigned based on its intensity, coordination, and via comparison to other SLC26 and SLC4 members, while the density at the S2 site was assigned based on its intensity and coordination. Both assignments are further supported by their location at the crossover region between the two positive helical dipoles from TM3 and TM10, and the proximity of the R409. Further functional studies increased our confidence in the assignments.

We have added the labels to helices in Fig. 3a-c.

Line 107 – please reference at least one of the ‘previous publications’.

Thank you for pointing out this shortcoming; we have added the references for publications that use the naming convention.

Line 117 – missing ‘for’.

Thank you for catching this error; it has been corrected in the revised manuscript.

Line 121-22. This sentence seems unnecessary. Would anyone have any reason to call into question the topology of Pendrin, given the sequence and structural similarity to other SLCs?

Agreed, the evidence should be clear for the topology of pendrin. This has been updated in the revised manuscript.

Line 136 – Can the interaction between F141 and the anion in site S1 strictly be called an anion- π ? I thought this term was for interactions with the delocalised electron cloud on the face of aromatics.

Cation- π interaction is perhaps more commonly seen in the structures and the cation is stabilized by the delocalized electrons on face of the aromatic ring, which is partially negative. Anion- π interactions is probably less common and is defined by interactions of an anion with the edge of the aromatic ring, which is partially positive due to delocalization of electrons to the faces of the ring.

Lines 139-144 – this section seems out of context. I suggest rewriting it to make it more straightforward for the reader. For example, the fact about R409H appears bolted on. Incidentally, in Pendred syndrome, is Pendrin correctly trafficked but non-functional, or do these mutations cause folding defects? Similarly, for Lines 147-149.

Thank you and agree. We have edited both sections in the revised manuscript.

Lines 163 – Do the mutations around the unknown ion binding site impact protein stability or expression? This data might inform on a structural function for the novel ion in the STAS domain. It would also be interesting to compare this region with the other STAS-containing structures. Is this feature unique to Pendrin?

The mutations to the binding site of an unknown ion, H698A and K702E, have lower levels of expression. The residues are not conserved in any of the SLC26 members. His698 is not conserved across species in pendrin, as the human version is Tyr.

Line 179 – I wondered whether there was any correlation between the severity of Pendrin syndrome and the location of the mutations. Indeed, are there levels of severity in the disease?

There have been reports of different levels of severity of Pendred syndrome, such as hearing impairment being bilateral or asymmetric for just one ear, and ranging from mild to profound. However, there is not enough information from databases (such as the NIH ClinVar database) to show whether there is a correlation between the severity of Pendred syndrome and the location of the mutations.

Line 207 – the authors suggest the mechanism of NFA inhibition is simply jamming the transporter in the inward open state. However, I'm left wondering whether NFA can bind the outward open state of the transporter and then clog in the inward-facing state or whether NFA enters the cell via another route and then jams Pendred. Could the authors please elaborate on how they think NFA enters the cell and interacts with Pendred? In addition, given that homologous structures of SLC4 and SLC26 structures have been determined in both outward and inward-facing states, do the authors think a structural comparison showing how NFA would hypothetically jam Pendred using a homology model of the outward-facing state in the SI (linked to the structural comparison in Fig. 7) might illustrate this point more clearly?

We appreciate this thoughtful question. We do not know if pendrin can transport NFA. Despite our best efforts, we were not able to identify a commercial source that offers radiolabeled NFA that would allow us to perform direct NFA transport measurements. Given the exorbitant costs for custom-synthesized radiolabeled compounds and the unfavorable lead times for its synthesis are also precluding factors of such an approach. The structure of SLC4A-NFA has been determined in the outward-facing conformation⁸. However, an outward-facing conformation of pendrin modeled based on the SLC4A structure shows that the NFA binding pocket is not preserved. We have added these comments in the Discussion.

References:

1. Nimigean, C. M. A radioactive uptake assay to measure ion transport across ion channel-containing liposomes. *Nat. Protoc.* **1**, 1207–1212 (2006).
2. Weng, J. *et al.* Insight into the mechanism of H⁺- coupled nucleobase transport. *Proc. Natl. Acad. Sci.* **120**, (2023).
3. Malinauskaite, L. *et al.* A mechanism for intracellular release of Na⁺ by neurotransmitter/sodium symporters. *Nat. Struct. Mol. Biol.* **21**, 1006–1012 (2014).
4. Soleimani, M. *et al.* Pendrin: An apical Cl⁻/OH⁻/HCO₃⁻ exchanger in the kidney cortex. *Am. J. Physiol. - Ren. Physiol.* **280**, 356–364 (2001).
5. Paulmichl, S. D. *et al.* Molecular and Functional Characterization of Human Pendrin and its Allelic Variants. *Cell. Physiol. Biochem.* **28**, 451–466 (2011).
6. Shi, L., Quick, M., Zhao, Y., Weinstein, H. & Javitch, J. A. The Mechanism of a Neurotransmitter:Sodium Symporter-Inward Release of Na⁺ and Substrate Is Triggered by Substrate in a Second Binding Site. *Mol. Cell* **30**, 667–677 (2008).
7. Lucas, X., Bauzá, A., Frontera, A. & Quiñonero, D. A thorough anion- π interaction study in biomolecules: On the importance of cooperativity effects. *Chem. Sci.* **7**, 1038–1050 (2016).
8. Capper, M. J. *et al.* Substrate binding and inhibition of the anion exchanger 1 transporter. *Nat. Struct. Mol. Biol.* (2023) doi:10.1038/s41594-023-01085-6.

Reviewers' Comments:

Reviewer #1:

Remarks to the Author:

I am more than satisfied with the response to my previous evaluation and have no further concerns.

Reviewer #2:

Remarks to the Author:

The authors have added several additional experiments to:

- more fully characterise the kinetics of ssPendrin in the liposome assays.
- support the identity of the novel S2 anion binding site using radioactive binding studies coupled with SDM.
- provided more context for the disease mutations and potential mechanism of NFA.

I am happy to support publication of this excellent study.